# RNA-Seq Analysis Reveals CCR5 as a Key Target for CRISPR Gene Editing to Regulate In Vivo NK Cell Trafficking

**DOI:** 10.3390/cancers13040872

**Published:** 2021-02-19

**Authors:** Emily R. Levy, Joseph A. Clara, Robert N. Reger, David S. J. Allan, Richard W. Childs

**Affiliations:** 1National Heart Lung and Blood Institute, National Institutes of Health, Bethesda, MD 20814, USA; emily.levy@nih.gov (E.R.L.); joseph.clara@nih.gov (J.A.C.); robert.reger@nih.gov (R.N.R.); david.allan@nih.gov (D.S.J.A.); 2The Department of Molecular Medicine, George Washington University, Washington, DC 20052, USA

**Keywords:** NK cells, cellular immunotherapy, chemotaxis, CRISPR, transcriptomics, gene-editing, lymphocyte homing

## Abstract

**Simple Summary:**

Adoptive immunotherapy utilizing ex vivo expanded natural killer (NK) cells is being explored in the clinical and preclinical settings to treat hematological tumors. Previous work has shown that a large fraction of ex vivo expanded NK cells traffic into the liver following i.v. infusion. In this manuscript, Levy et al. show that ex vivo expansion of NK cells alters the mRNA transcription and surface expression of several chemokine receptors. The observed shift in chemotactic receptor expression may compromise the homing of infused cells into sites where hematological tumors reside, such as bone marrow, lymph nodes, and peripheral blood, by promoting preferential trafficking into liver tissue. Here we demonstrate clustered regularly interspaced short palindromic repeats (CRISPR) gene abrogation of C-C chemokine receptor type 5 (CCR5) as a novel strategy that reduces the trafficking of adoptively transferred ex vivo expanded NK cells into liver tissue and increases NK cell presence in the circulation.

**Abstract:**

A growing number of natural killer (NK) cell-based immunotherapy trials utilize ex vivo expansion to grow and activate allogenic and autologous NK cells prior to administration to patients with malignancies. Recent data in both murine and macaque models have shown that adoptively infused ex vivo expanded NK cells have extensive trafficking into liver tissue, with relatively low levels of homing to other sites where tumors often reside, such as the bone marrow or lymph nodes. Here, we evaluated gene and surface expression of molecules involved in cellular chemotaxis in freshly isolated human NK cells compared with NK cells expanded ex vivo using two different feeder cells lines: Epstein-Barr virus (EBV)-transformed lymphoblastoid cell lines (LCLs) or K562 cells with membrane-bound (mb) 4-1BB ligand and interleukin (IL)-21. Expanded NK cells had altered expression in a number of genes that encode chemotactic ligands and chemotactic receptors that impact chemoattraction and chemotaxis. Most notably, we observed drastic downregulation of C-X-C chemokine receptor type 4 (CXCR4) and upregulation of C-C chemokine receptor type 5 (CCR5) transcription and phenotypic expression. clustered regularly interspaced short palindromic repeats (CRISPR) gene editing of CCR5 in expanded NK cells reduced cell trafficking into liver tissue and increased NK cell presence in the circulation following infusion into immunodeficient mice. The findings reported here show that ex vivo expansion alters multiple factors that govern NK cell homing and define a novel approach using CRISPR gene editing that reduces sequestration of NK cells by the liver.

## 1. Introduction

Clinical trials targeting both hematologic malignancies and solid tumors have shown that adoptive therapy with allogeneic and autologous natural killer (NK) cells is safe, with clinically meaningful tumor responses being observed in multiple studies [1,2,3]. Ex vivo activation and expansion of primary NK cells through contact with irradiated feeder cells is commonly utilized to produce large numbers of highly cytotoxic NK cells for clinical use. [4,5,6]. Irradiated Epstein-Barr virus (EBV)- lymphoblastoid cell lines (LCLs)and genetically engineered K562 (GE-K562) cells are two feeder cell populations commonly used to expand NK cells for clinical use. Modifications to the K562 cell line to promote NK cell expansion ex vivo include induced expression of membrane-bound (mb) interleukin (IL)-15 or IL-21, along with the costimulatory molecule 41BB ligand [7,8,9,10]. Although numerous clinical and pre-clinical investigations have focused on improving NK cell in vivo persistence and cytotoxicity, relatively few studies have focused on optimizing properties of NK cells that impact their capacity to traffic with specificity within the host to niches where hematological or solid malignancies reside. 

NK cells can be mobilized to sites of inflammation through signaling by way of chemokines. The four classifications of chemokine domains are CC, CXC, CX3C, and XC. The nomenclature of the protein depends on the configuration of the two cysteine residues most proximal to the N-terminus [11]. Cell surface chemokine receptors are denoted by “R,” and chemokine ligands are denoted by “L.” The major chemokine receptor-ligand pairs that modulate the trafficking of peripheral blood NK cells include CCR1-CCL3/CCL5, CCR5-CCL3/CCL5, CCR7-CCL19/CCL21, CXCR3-CXCL9/CXCL10, and CXCR4-CXCL12 (also known as stromal derived factor-1 alpha (SDF-1α) [12,13,14]. Cell trafficking can also be influenced by other chemoattractant molecules, such as S1P and chemerin, by signaling through sphingosine-1-phosphate receptor 5 and chemerin chemokine-like receptor 1 (CMKLR1), respectively [15,16]. NK cell expression of chemokine receptors and other molecules that are important for cell trafficking evolve through cell development due to different environmental cues. Additionally, as a mechanism to link innate and adaptive immune responses, NK cells produce various chemokines and cytokines, such as CCL3, CCL5, granulocyte-macrophage colony-stimulating factor (GM-CSF), interferon gamma (IFN-γ), and tumor necrosis factor- 1 alpha (TNF-α), to recruit and activate alternate immune cells to sites of infection or the tumor microenvironment [17]. 

Accordingly, it is important to consider that the therapeutic efficacy of NK cell-based therapies may ultimately depend on their ability to mobilize to target tissues following adoptive transfer. Although our current understanding of chemotactic and chemoattractant properties of ex vivo expanded NK cells is limited, recent data have shown that NK cell trafficking into environments where hematological malignancies reside, such as the lymph nodes and bone marrow, may be limited followed their adoptive transfer into patients [18]. Therefore, the objective of this study was to elucidate the phenotypic and transcriptional features of NK cells that have been expanded ex vivo with feeder cells, focusing on chemotactic molecules that play a role in cellular trafficking. Our analysis revealed that expanded NK cells had substantial alterations in multiple chemokine receptors compared to fresh NK cells. The pattern of gene expression alterations we observed would likely direct adoptively transferred NK cells to the liver, potentially impeding their trafficking into other tissue sites harboring tumor. Importantly, using these data, we show for the first time that clustered regularly interspaced short palindromic repeats (CRISPR) gene editing of a chemokine receptor (CCR5) can redirect NK cell tracking in vivo. This discovery lays the foundation for a strategy that has the potential to bolster the effectiveness of adoptive NK cell immunotherapy for both hematological and solid tumors.

## 2. Materials and Methods

### 2.1. Primary Cells and Cell Lines

Peripheral blood mononuclear cells (PBMCs) from healthy donors were isolated using Lymphocyte Separation Medium (MP Biomedicals, Irvine, CA, USA) (NIH protocol 99-H-0050 and 2006/229–31/3. NK cells were isolated by magnetic bead separation; cluster of differentiation (CD)3-depleted and CD56-selected (Miltenyi Biotech, Germany) and expanded in vitro for 14–16 days in G-Rex (Wilson Wolf Manufacturing, St Paul, MN, USA) flasks with irradiated human EBV-LCL feeder cells at a ratio of 1:10 or with GE-K562 cells containing membrane bound IL-21 and 41BB ligand (K562. mbIL-21.41BBL) at a ratio of 1:2. SMI-EBV-LCLs were established for Dr. Richard Childs’ lab by The Production Assistance for Cellular Therapies group (NIH/NHLBI, Bethesda, MD), and K562.mbIL-21.41BBL was generously provided by Dr. Katy Rezvani from MD Anderson Cancer Center (Houston, TX). EBV-LCLs were authenticated in 2009 by The Center for Cell-based Therapy (NIH/CCR, Bethesda, MD). Authentication for the genetically modified K562 cell line was not performed upon being received. Cell lines were propagated in Roswell Park Memorial Institute (RPMI)-1640 supplemented with 10% heat-inactivated fetal bovine serum (FBS) (Sigma-Aldrich, Germany). NK cells were cultured in X-vivo 20 media (Lonza, Switzerland) containing 10% human AB serum (Sigma-Aldrich), 1% Glutamax (Gibco, Gaithersburg, MD, USA), and 500 U/mL IL-2 (Tecin^TM^, NIH/NCI, Fredrick, MD). For NK cell expansion, fresh media was supplied to cells starting on day 5 and then every 2–3 days until the cells were harvested for use in experiments. During NK cell expansion cells were maintained to have a consistent cell concentration of 0.8–1.5 × 10^6^/mL between all samples. 

### 2.2. Flow Cytometry 

Directly following NK cell isolation from healthy donor PBMCs (*n* = 6), population purity was defined by flow cytometry using the following panel: CD56-PE Cy7 (BD biosciences, clone NCAM16), CD3-FITC (BD Biosciences, San Jose, CA, USA, clone SK7), CD19-PE (BD Biosciences, clone SJ25C1), and CD14-PB (Biolegend, San Diego, CA, clone M5E2). Freshly isolated NK cells from each donor that were rested overnight in NK media without IL-2 and paired samples that were expanded for 16 days ex vivo were stained using the following antibody reagents: CXCR6 (BD Biosciences, clone 13B 1E5), CCR1 (Biolegend, clone 5F10B29), CCR5 (Biolegend, clone J418F1), CD151 (BD Biosciences, clone 14A2.H1), CD11A (BD Biosciences, clone HI111), CD49D (Biolegend 9F10), CD2 (BD Biosciences, clone S5.2), DNAM1 (Biolegend, clone 10E5), CD18 (Biolegend, clone CBR LFA 1/2), CD9 (BD Biosciences, clone M-L13), CD29 (Biolegend, clone TS2/16), CCR6 (BD Biosciences, clone 11A9), CXCR3 (Biolegend, clone G025H7), CD44 (Biolegend, clone IM7), PSGL1 (Biolegend, clone KPL-1), CXCR4 (Biolegend, clone 12G5), CCR7 (Biolegend, clone 4B12), CXCR1 (Biolegend, clone 8F1), CX3CR1 (Biolegend, clone 2A0-1). Following 30 min of antibody staining at 4 °C, cells were washed with Phosphate-buffered saline (PBS) containing 10% FBS and 2 mM Ethylenediaminetetraacetic acid (EDTA) and fixed with PBS containing 1% paraformaldehyde. Flow cytometry was conducted using an LSRFortessa cytometer (BD biosciences) and data was analyzed using FlowJo (BD biosciences, version 9)

### 2.3. RNA-Sequencing and Analysis

RNA was isolated from 1 × 10^7^ fresh NK cells and 1 × 10^7^ ex vivo expanded NK cells after 16 days of culture with RNAeasy minikit (Qiagen, Germany). RNA isolates were submitted to Novogene (Sacramento, CA, USA) for sequencing and analysis. Sequencing libraries were prepared, and quality of the libraries was assessed by Qubit (Thermo Fischer Scientific, Wilmington, DE, USA), 2100 Bioanalyzer (Aligent, Santa Clara, CA, USA), and qPCR. Qualifying samples were sequenced by HiSeq (Illumina, San Diego CA, USA), and 150-bp paired-end reads were generated at a depth of 20×. Sequencing-generated fastq files were checked for read quality and mapping rate with FastQC (version 0.11.8), clean reads were aligned with STAR (version 2.5) to hg38 reference genome. Gene expression level was determined using HTSeq (version 0.6.1) and DESeq2 R package (version 2_1.6.3). Data is reported as Fragments Per Kilobase of exon model per Million mapped reads (FPKM).

### 2.4. Identification of Soluble Factors in Ex Vivo NK Cell Expansions

Supernatants were collected from NK expansion cultures as described above on days 2, 3, 4, and 5 and stored at −20 °C. Concentrations of analytes were measured using a Mesoscale Discovery multiplex panel (Mesoscale Diagnostics). Supernatants were thawed and assessed for the presence of IFN-γ, TNF-α, CCL3, IL-10, vascular endothelial growth factor (VEGF), GM-CSF, SDF-1α, with IL-2 used as a positive control.

### 2.5. Cytokine Exposure Assay

NK cells were freshly isolated from healthy donor PBMCs (*n* = 3 donors) using Rosette Sep (Stem Cell Technologies, Vancouver, BC, Canada). Fresh NK cells were resuspended in NK media without IL-2 at a concentration of 5 × 10^5^ cells/mL in a 24-well cell culture plate (Corning). Next, 20 ng/mL rh (recombinant human) G-CSF (Amgen, Thousand Oaks, CA, USA), 200 U/mL IL-2, 50 ng/mL rhTNF-α (R&D Systems, Minneapolis, MN, USA), 200 ng/mL rhIFN-γ (R&D Systems), 20 ng/mL rhIL-4(R&D Systems), 20 ng/mL rhIL-15(R&D Systems), 20 ng/mL rhGM-CSF (R&D Systems), 20 ng/mL rhIL-10 (R&D Systems), 5 ng/mL rhTGF-β (R&D Systems), 50 ng/mL rhCCL5 (R&D Systems), 50 ng/mL rhCCL3 (R&D Systems), 50 ng/mL rhVEGF165 (R&D Systems), or 100 ng/mL rhSDF-1α (R&D Systems) were individually added to each well. NK cells in media with no cytokine were used as the control. All samples were incubated for 24 h in 37 °C 6.5% CO_2_. Cells from each sample were collected and stained for flow cytometry-based identification of CCR1, CCR5, CXCR4, and CXCR6 surface expression. 

### 2.6. In Vivo Trafficking of CCR5 Knockout (KO) NK Cells

Gene KO Kit v2 (Synthego, Redwood City, CA, USA) was used to specifically target CCR5 in primary NK cells. Primary NK cells were expanded ex vivo using irradiated SMI-EBV-LCL feeder cells as described above. NK cells were collected on day 7 and were electroporated (Maxcyte ATX) with CRISPR/Cas9 ribonucleoprotein (RNP) complexes. Three synthetic guide RNAs (sgRNAs) were used in a single electroporation: UUUUGCAGUUUAUCAGGAUG, AAAACAGGUCAGAGAUGGCC, UGUAUUUCCAAAGUCCCACU, with each sequence preceding a Synthego modified EZ scaffold. Following CRISPR/Cas9 KO of CCR5, NK cells were maintained in expansion culture. On day 17, CCR5 KO NK cells and non-modified NK cells were collected, rinsed 2x with PBS and suspended in PBS for i.v. infusion into 6 -month-old NOD.Cg-Prkdc^scid^Il2rg^tm1Wjl^/SzJ (NSG)mice (Jackson Laboratories, Bar Harbor, ME). Mice received 1 × 10^7^ CCR5 KO NK cells (*n* = 4) or non-modified NK cells (*n* = 3), with 1 × 10^5^ U IL-2, and then were harvested 24 h following injection; blood, bone marrow, and livers were collected. Cells were manually dissociated and evaluated for viability and CD56; 100 uL of each sample was acquired with an LSR Fortessa cytometer and the data was analyzed with FlowJo. Cells were gated based on size and then single, Livedead-, CD56+ events were acquired from each mouse. The percentage of NK cells within each organ was determined as a fraction of all of the NK cells acquired per mouse across each organ (*n* = 8–9 per group, 2 independent experiments). 

### 2.7. Analysis of Statistics

Statistical analysis of RNA-seq data was output by DESeq2 R package (2–1.6.3). Benjamini and Hochberg’s false discovery rate approach was utilized to calculate the adjusted *p* values for each event. Genes with an adjusted *p* value of <0.05 found by were determined to be differentially expressed. Principle component analysis was performed using the Bioconductor package pcaExplorer (v2.16.0), as described in Marini et al (2019) [19]. All other data were analyzed with PRISM 7.0b software (GraphPad Prism Inc., San Deigo, CA, USA), using the two-tailed paired *t*-test, Student’s *t*-test, the Welch’s-*t*-test to report significant differences between groups. The appropriate test was chosen based on data distribution, variances, and experimental set up. * *p* < 0.05, ** *p* < 0.01, **** *p* < 0.0001, and ns = not significant.

## 3. Results

### 3.1. RNA Sequencing Reveals That NK Cells Expanded Ex Vivo With Feeder Cells Have a Vastly Different Transcriptional Landscape Compared to NK Cells Freshly Isolated from Peripheral Blood

Ex vivo activation and expansion of primary NK cells using irradiated feeder cells is commonly utilized to produce large numbers of highly cytotoxic NK cells for clinical use. Therefore, we performed a detailed analysis using flow cytometry and RNA-seq to characterize phenotypic and gene transcription changes associated with ex vivo expansion of NK cells using two feeder lines utilized for clinical use: K562.mbIL-21.41BBL and EBV-LCLs (Appendix A). The majority of freshly isolated NK cells and all of the ex vivo expanded NK cell samples utilized in this study were confirmed to be a highly viable and purified CD56+ NK cell population devoid of T-cells (less than 0.2% T cell contamination, *n* = 6 HDs) (Appendix A).

As determined by FastQC quality check analysis of the sequenced samples, >94% of samples achieved a quality score of >30, and all samples achieved a mapping rate of >95%. Principle component analysis (PCA) and an unsupervised heat map clustering analysis of ex vivo expanded and fresh NK cell samples exhibited distinct gene expression properties of these cell populations (Figure 1A,B). PC1 accounts for 90% of the transcriptional differences between all of the samples and represents the effects of ex vivo expansion. PC2 accounts for approximately 5% of differences that were exhibited amongst the fresh NK cell samples and is likely due to donor-specific genetic variability; all other components account for less than 1% of variance (Appendix A). 

Compared to fresh NK cells, ex vivo expansion of NK cells with EBV-LCL cells led to an upregulation of 4257 genes and a downregulation of 7661 genes, using an adjusted *p* value threshold of <0.05 assigned by DESeq2 algorithm. In a similar fashion, ex vivo expansion of NK cells using GE-K562 led to an upregulation of 4981 and a downregulation of 8151 genes (Figure 1C). The populations of NK cells expanded with either feeder cell line were transcriptionally similar; significant differential gene expression that was observed included the upregulation of 13 genes and the downregulation of 10 genes, with each of these genes having low transcript abundance (FPKM < 30) (Appendix A). Gene ontology enrichment analysis revealed that the most enriched differentially expressed gene sets in ex vivo expanded NK cells compared to fresh NK cells were genes involved with mitosis and DNA replication pathways (Figure 1D).

Transcription for genes that are critical for NK cell cytotoxicity was significantly higher in ex vivo expanded NK cells, such as GZMB, GZMA, PRF1, FAS, FASL, TNFSF10, 2B4, NCR1, NCR3 (Figure 2A,B). Although many of the functional and phenotypic features we observed on expanded NK cells have previously been reported, this is the first report to characterize the vast scope of changes associated with ex vivo expansion on a transcriptional level using RNA-sequencing. 

### 3.2. RNA-Sequencing of Fresh and Ex Vivo Expanded NK Cells Reveals Strong Transcriptional Shifts of Genes That Control Cell Trafficking 

Given the homing of adoptively transferred NK cells to sites of malignancy is likely an important determinant of therapeutic efficacy, we focused our efforts to detail differential gene expression of chemotactic and adhesion molecules known to regulate cell trafficking. Among the largest and most statistically significant differences observed in our dataset were the downregulation of CXCR4 and CCR7, and the upregulation of CXCR3, CXCR6, CCR5, and CCR1 in expanded NK cells compared to fresh NK cells (Figure 3A,B). Specifically, CXCR4 was downregulated >10-fold (adjusted *p* = 1.14 × 10^−33^) and CXCR6 was upregulated approximately 60-fold (adjusted *p* = 3.00 × 10^−31^) on LCL-expanded NK cells. Transcription of CCR5 and CCR1 were also substantially upregulated in expanded NK cells compared to fresh NK cells (>50-fold, adjusted *p* = 8.76 × 10^−30^ and >4-fold, adjusted *p* = 9.3 × 10^−5^, respectively). Remarkably, we also found a substantial increase in transcription of chemokine ligands XCL1, XCL2, CCL3, CCL4, and CCL5, and a significant loss in transcription of CXCL8, CXCL16, and SIPR5 (Figure 3A,B). 

Among adhesion molecules, the most notable changes in transcription occurring with ex vivo NK cell expansion was a decrease in CD62L (approximately 5-fold, adjusted *p* = 1.82 × 10^−11^) and a strong increase in CD2 (approximately 4-fold, *p* = 4.0 × 10^−11^) (Figure 3A,B). Genes that are responsible for the downstream signaling of chemokine receptors, including genes involved in mitogen-activated protein (MAP), Rho-associated, and phosphoinositide-3 (PI3) kinase signal cascades, remained unchanged (Appendix A). We also assessed changes in gene expression of CXCR4 transcriptional regulators and post translational regulators. Of these genes, PIM-1 and GRK6 were increased in expanded NK cells, NRF-1, and YY-1 gene transcription were reduced in expanded NK cells, and CREB-1 expression was unchanged in expanded NK cells compared to fresh NK cells (Figure 3C).

### 3.3. Ex Vivo Expansion With Feeder Cells Induces a Shift of Surface Chemotactic Receptor Expression on NK Cells 

To support our findings of transcriptional alterations, we profiled the surface expression of critical lymphocyte homing molecules on the surface of fresh and ex vivo expanded NK cells. Consistent with previous reports on the effects of ex vivo expansion on NK cell phenotype, our RNA-seq data showing downregulated CXCR4 transcription corroborated with reduced CXCR4 surface expression observed on expanded NK cells (Figure 3A,B). Surface expression of CCR7, CXCR1, and CX3CR1 all also significantly decreased following ex vivo expansion. In contrast, integrins, such as CD11a, b, and c, as well as CD18, CD29, and CD49, all increased on the surface of NK cells (Figure 4A). Likewise, tetraspanins (plasma membrane organizers that enhance cell adhesion and trafficking), including CD151 and CD9, were also upregulated following expansion. Together, these data suggest that expanded NK cells should have an increased propensity to firmly adhere to tissue endothelium. Compared to the fresh NK cell controls, the chemokine receptors CXCR6, CCR1, CCR5, and CXCR3 increased substantially after expansion. Notably, -LCL-expanded NK cells showed significantly higher levels of surface CCR1 and CCR5 compared to GE-K562 expanded NK cells (Figure 4A,B). These chemokine receptors have previously been shown to promote NK cell trafficking from the periphery into the liver during viral infection [20,21]. CXCR6, along with CCR5, is involved in retention of liver-resident NK cells within hepatic sinusoids [22,23]. Taken together, ex vivo expansion of NK cells with feeder cells and stimulatory cytokines leads to alterations in genes regulating chemokine receptor surface expression, resulting in a phenotypic shift that might be expected to promote NK cell trafficking to the liver and lessen the capacity for NK cells to travel to the bone marrow (BM).

### 3.4. Soluble Factors Present Within Ex Vivo NK Cell Expansion Cultures May Regulate CXCR4 Expression

To elucidate the possible mechanisms inducing changes in chemokine receptor phenotype during ex vivo expansion, we investigated the role of soluble factors present in culture. We compared soluble factors in cultures containing NK cells alone in media with IL-2 media versus NK cell cultures stimulated with irradiated LCL feeder cells in the same media. Supernatants collected on days 2, 3, 4, and 5 were analyzed using a multiplex analysis to quantify the presence of GM-CSF, IL-10, IL4, SDF-1α, TNF-α, VEGF, and CCL3, as these factors have previously been identified as possible regulators of chemokine receptor expression on various cell types. In contrast to NK cells in IL-2 media alone, supernatants from NK cells cultured with irradiated LCL feeders showed increasing concentrations of GM-CSF, TNF-α, and VEGF over time (Figure 5A). Conversely, the concentration of IL-10, which was increased compared to NK cells stimulated with media, decreased over time, while a sustained elevation was seen with SDF-1α (Figure 5A). 

Next, we examined the role that each of the above soluble factors played in the regulation of chemotactic receptor expression. We cultured fresh NK cells from healthy donor PBMCs in NK media with various cytokines in vitro for 24 h. We observed that VEGF, TNF-α, IL10, IL4, IFN-γ, GM-CSF, GCSF, CCL5, and CCL3 had no direct effect on the surface expression of CXCR4, CXCR6, CCR5, or CCR1 (Figure 5B). In contrast, there was a slight increase in CXCR4 expression 24 h after NK cells were exposed to TGF-β, and a substantial decrease in CXCR4 expression 24 h after NK cells were exposed to SDF1a, IL-15, and IL-2 (Figure 5B). The above findings were in accordance with previously published data [24,25,26].

### 3.5. CRISPR/CAS9 Gene Editing of CCR5 Redirects NK Cell Trafficking In Vivo Following Adoptive Transfer Into Mice

As our data show, ex vivo expanded NK cells exhibit a striking upregulation of CCR5, a chemokine receptor associated with trafficking to and retention in the liver. Therefore, we evaluated if CRISPR gene editing of this chemokine receptor could be used as a strategy to redirect NK cell trafficking in vivo in immunodeficient mice receiving infusions of ex vivo expanded human NK cells. Seven days following ex vivo expansion, NK cells were electroporated with a mix of 3 sgRNAs targeting CCR5 complexed with Cas9. Electroporated cultures harvested on day 14 maintained their proliferative capacity and had substantially reduced CCR5 expression (compared to non-electroporated control NK cells (Figure 6A,B). Both expanded NK cell populations were then injected i.v., into NSG mice, with blood, BM, lungs, and livers being harvested 24 h following infusion. Disrupting CCR5 in NK cells significantly reduced cellular trafficking into the liver compared to control NK cells (Figure 6C). Moreover, there were significantly more CCR5 CRISPR/Cas9 disrupted NK cells in the circulation and in the lung compared to control NK cells. These data suggest that the upregulation CCR5 in expanded NK cells promotes NK cell trafficking from the circulation into the liver tissue following IV infusion. 

## 4. Discussion

In vitro co-culture of NK cells with feeder cells is commonly used as an approach to generate large numbers of NK cells for use in clinical trials [7,27,28]. For NK cells, these cultures create an artificial environment designed to optimize their cytotoxicity and prolific expansion. Although culturing NK cells is known to result in phenotypic and functional changes, the transcriptional impact of ex vivo expanding NK cells on chemotactic molecules has not previously been characterized. The first aim of our study was to provide the first comprehensive analysis comparing ex vivo expanded NK cell populations on a transcriptional and phenotypic level to resting NK cells. This work identified that the transcription of chemoattractant ligands and expression of chemotactic receptors are substantially altered as a consequence of ex vivo expansion. We have previously shown with macaque and murine models that after i.v. infusion of expanded NK cells, the majority of infused NK cells are taken up by liver tissue [18,25]; therefore, the second aim of our study was to mitigate this sequestration by disrupting CCR5 in expanded NK cells by CRISPR gene editing. 

Our RNA-seq data shows that the process of ex vivo expansion using two different feeder cell populations in IL-2 containing media results in the differential expression of more than ten thousand genes. Genes involved in the regulation of cell cycle, DNA replication, and mitotic spindle organization were among those with significant differential expression. This finding is consistent with the notion that NK cell expansion cultures induce strong proliferative effects. From a transcriptional standpoint, LCL-expanded and K562-expanded NK cells were remarkably similar, with only thirteen genes having significant differential expression. Foundational studies have established that ex vivo expansion alters numerous NK cell surface receptors and augments NK cell cytotoxicity. Our transcriptome analysis provides a far greater scope of analysis than has previously been reported, which both corroborates and extends prior findings. We observed significant gene-upregulation of many activating receptors and cytotoxicity effector molecules, including NKp46, and NKp30, FASL, and TRAIL. Likewise, RNA transcription was augmented for cytotoxic granules involved in mediating NK cell cytotoxicity, such as granzyme A and B, as well as perforin. Although many of the functional and phenotypic changes observed in expanded NK cells have previously been reported, our study describes these changes in the context of a global transcriptional analysis using RNA-seq. 

In addition to effects on NK cell activation and cytotoxicity, RNA-seq data showed NK cells expanded ex vivo upregulated chemotactic ligands that are implicated in bridging innate and adaptive immune responses, including XCL1, XCL2, CCL5, CXCL16, and CXCL8. NK cell secretion of XCL1 and XCL2 facilitates dendritic cell presentation of antigens and priming of cytotoxic T cells [29,30,31]. Expression of these genes, as well as CCL5, a chemoattractant for both dendritic cells and T cells, was significantly increased in NK cells after ex vivo expansion. These changes may enhance the ability of expanded NKs to promote antitumor activity against melanoma, breast cancer, lung cancer, and head and neck squamous cell carcinoma [32]. Existing data confirms that NK cells can directly enhance adaptive immunity by priming antitumor CD8+ T cell responses in the tumor microenvironment via IFN-γ [33]. Thus, the changes that we observed in ex vivo expanded NK cells could bolster their ability to orchestrate adaptive antitumor immune responses. 

Moreover, our analysis revealed that ex vivo expansion of NK cells led to a substantial reduction in the transcription of CXCL8, CXCL16, and CMKLR1. These alterations could have an impact on multiple arms of the human immune system given that (a) CXCL8 plays a role in attracting neutrophils to sites of inflammation, (b) CXCL16 attracts CXCR6-expressing dendritic cells and T cells and, and (c) CMKLR1, the chemotactic receptor for chemerin, facilitates the co-localization of NK cells with DCs at sites of inflammation. It should be noted that Parolini et al. previously showed that 24-h in vitro culture of NK cells with IL-2 and IL-15 led to similar reductions in CMKLR1 mRNA as observed in our study of NK cells undergoing ex vivo expansion with feeder cells [16]. These findings provide a rationale to further investigate the full functional consequences of altered expression of CXCL8, CXCL16, and CMKLR1 on NK cell antitumor immunity. 

Our data also reveal that ex vivo expansion induces profound transcriptional shifts in genes that control cell trafficking. Specifically, we found downregulation of CCR7, CXCR2, and CXCR4 and upregulation of CCR1, CCR5, CCR8, CXCR3, and CXCR6. Consequently, the ability of adoptively transferred NK cells to target an array of tumors types is limited. Pertinent examples of the importance in understanding and manipulating chemokine receptor expression on therapeutic NK cell products can be demonstrated in studies evaluating CXCR3. Ponzetta et al. (2015) showed that multiple myeloma patients have elevated levels of CXCL9 and CXCL10 in the peripheral blood, which appears to cause endogenous NK cells to migrate away from the BM via a CXCR3-chemokine ligand gradient [34]. Notably, this chemokine receptor mediated deleterious effect on NK cell function can be overcome with CXCR3 blockade on adoptively transferred NK cells [35]. However, it is also important to consider that CXCR3 expression on ex vivo cultivated NK cells may offer an advantage in targeting cancers with elevated CXCL10 in the tumor microenvironment [36]. 

Like CXCR3, there are also profound functional implications of CXCR4 expression on NK cell products. We observed that NK cell exposure to molecules that are released into feeder cell co-culture supernatants, such as IL-2, IL-15, and SDF-1a, can contribute to a reduction in their surface expression of CXCR4. Therefore, the exogenous administration of IL-2 or IL-15 following the infusion of ex vivo expanded NK cells may further dampen their CXCR4 expression, which could substantially limit their ability to migrate into BM niches. Further, our study illuminates how transcription and surface expression of CXCR4 may be regulated. Our transcriptome analysis revealed substantial changes in CD62L, JAK2, PIM-1, GRK6, HIF-1α, YY-1, NRF-1, and CREB-1 gene expression. These genes have all independently been reported to regulate mRNA transcription, lymphocyte expression, and signaling function of CXCR4 [37,38,39,40,41,42,43,44,45]. Specifically, our data show an increase in PIM-1 (negative transcriptional regulator) and a decrease in NRF-1 and HIF-1α (positive transcriptional regulators) after ex vivo expansion. Thus, our data identifies these genes as potential novel targets for gene editing to modulate CXCR4 expression on expanded NK cells. 

Augmentation of surface CXCR4 expression results in improved bone marrow homing of adoptively transferred ex vivo expanded NK cells in a murine model [25]; however, with this model, the majority of infused CXCR4-modified cells were still sequestered by the liver. We believe that the upregulation of CCR5 is a significant contributor to the biodistribution of these infused cells. CRISPR-based gene editing has proven to be an efficient way to stably modify ex vivo expanded NK cells [46]; thus, we utilized CRISPR gene editing of CCR5 to overcome the “NK cell sponge” effect of the liver. CRISPR gene edited NK cells had substantially reduced CCR5 expression (26% CCR5^+^) compared to non-electroporated control NK cells (87% CCR5^+^). Twenty-four hours after infusion into NSG mice, CCR5 CRISPR-disrupted NK cells maintained their proliferative capacity and had significantly reduced cellular trafficking into the liver compared to control NK cells. Notably, a higher percentage of CCR5 CRISPR disrupted NK cells were observed in the circulation and in the lungs, confirming that CCR5 upregulation in expanded NK cells has a functional impact on their homing profiles. Future studies will evaluate the potential of pharmacological inhibition of CCR5 to reduce the trafficking of infused cells into liver tissue. 

Although adoptively transferred immune cells accumulate in the liver, it is not clear if they would be effective at controlling hepatic tumors. The liver tumor microenvironment is known to be particularly suppressive to endogenous hepatic NK cells, and would be predicted to have the same suppressive effects to adoptively transferred ex vivo expanded NK cells. Factors that contribute to the NK cell suppression by liver tumors include hypoxia, tumor production of metabolites, such as lactate and adenosine, tumor secretion of IL-10 and TGF-β, and Indoleamine 2, 3-dioxygenase and Prostaglandin-E_2_ released by tumor-derived fibroblasts [47,48,49]. These factors are characteristic of hepatic tumors that can induce NK cell dysfunction or anergy. 

Based on these data, future studies will explore whether deletion of CCR5 in combination with boosting surface CXCR4 can be used as a strategy to maximize the homing of adoptively transferred ex vivo expanded NK cells in hosts with bone marrow resident tumors. Such a strategy could potentially mitigate the issue of NK cell homing into the liver, and, additionally, redirect cellular trafficking into targeted tumor sites. Future studies will also evaluate how CCR1 and CXCR6 expression influence the biodistribution of infused expanded NK cells.

## 5. Conclusions

Ex vivo NK cell expansion using feeder cells is commonly used in clinical trials to grow and activate allogenic and autologous NK cell products prior to infusion into patients. Here, we report that the genomic landscape of NK cells is influenced dramatically by this process. Remarkably, we identified ex vivo expanded NK cells to have substantial alterations in expression of a number of genes that impact in vivo cellular trafficking, including CCR5, CCR1, CXCR3, and CXCR4. We go on to show that CRISPR-gene editing can be utilized to disrupt upregulated CCR5, and we further show that the homing of these gene-edited NK cells is redirected from the liver into the circulation following their infusion into immunodeficient mice. This work defines transcriptional alterations occurring in ex vivo expanded NK cells that may have clinical relevance and defines a novel strategy to redirect NK cell trafficking in vivo. Such an approach has the potential to improve the effectiveness of NK cell immunotherapy for hematological malignancies and other cancers.

## 6. Patents

An application for an intellectual property patent for CRISPR Cas9 gene editing of NK cell chemokine receptors has been filed. 

## Figures and Tables

**Figure 1 cancers-13-00872-f001:**
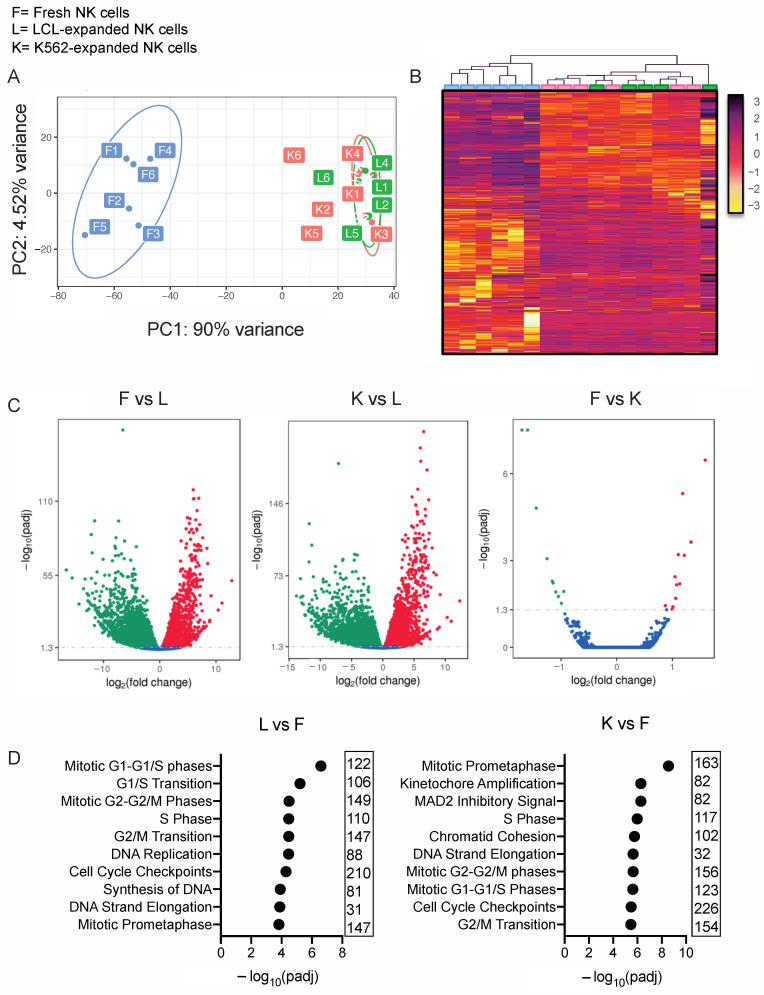
Ex vivo expansion heavily alters the natural killer (NK) cell transcriptional landscape. F = freshly isolated and rested NK cells, L = lymphoblastoid cell line (LCL)-expanded NK cells and K= genetically engineered K562 (GE-K562)-expanded NK cells. (**A**) Principal component analysis of fresh and expanded NK cells. (**B**) Unsupervised gene clustering of fresh and expanded NK cell populations (color scale represents relative gene expression). (**C**) Volcano plots to illustrate the distribution of significance (−log_10_(adjusted *p* value)) and change in expression (log_2_(fold change)) of differentially expressed genes within the comparisons. (**D**) Gene ontology enrichment analysis of top significant gene groups that are differentially expressed.

**Figure 2 cancers-13-00872-f002:**
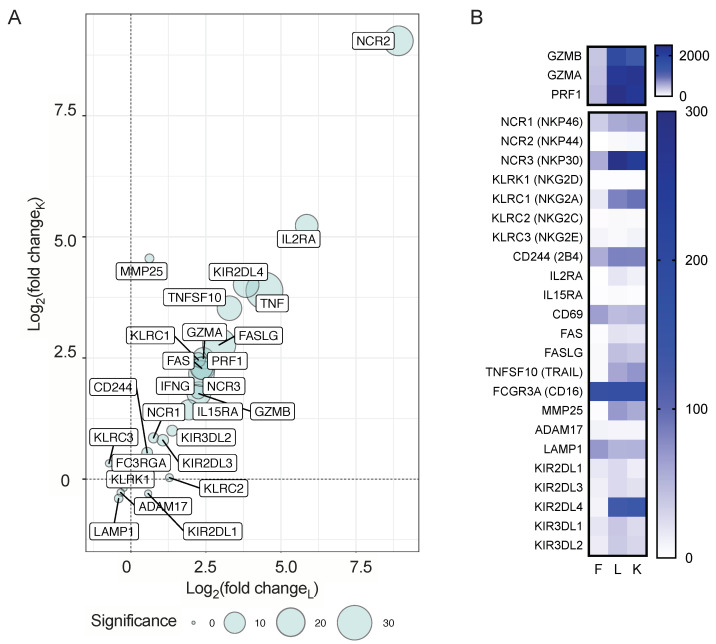
Ex vivo expansion of NK cells leads to a highly significant transcriptional alterations in genes that are crucial for NK cell cytotoxicity. (**A**) Bubble plot that displays genes that control NK cell cytotoxicity. Axis values represent log_2_-transformed fold change of transcript expression in expanded NK cells compared to fresh NK cells, where L = LCL-expanded NK cells and K = GE-K562-expanded NK cells. The size of the bubble corresponds to the log_10_-transformed adjusted *p* value to denote the statistical significance of differential expression, based on LCL-expanded NK cell values. (**B**) Heat maps that display Fragments Per Kilobase of exon model per Million mapped reads (FPKM) values of the genes that are graphed in (**A**).

**Figure 3 cancers-13-00872-f003:**
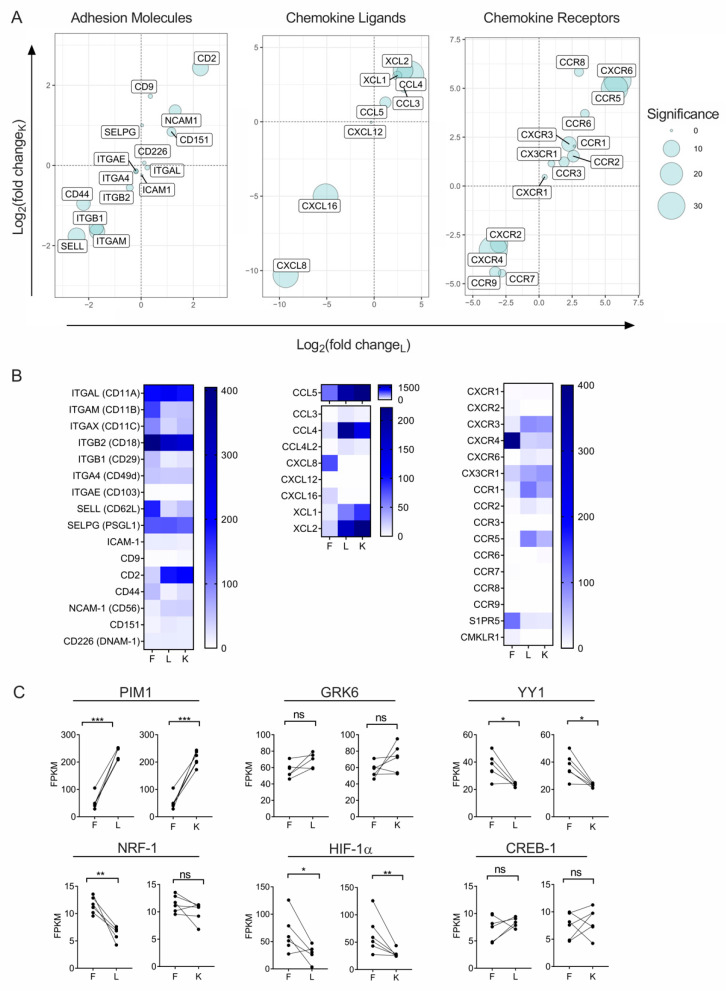
Ex vivo expansion of NK cells leads to a highly significant transcriptional alterations in genes that govern cellular trafficking. (**A**) Bubble plots that displays genes that control NK in vivo trafficking; Graphical organization is the same as in Figure 2A. (**B**) Heat maps that report FPKM values for the genes that are graphed in (A). (**C**) Line graphs showing the change in transcription of genes that regulate CXCR4 in NK cells from fresh (F) to expanded with either LCL feeder cells (L) or GE-K562 feeder cells (K), from individual donors. The paired *t*-test was used to determine significance, ns = not significant, * *p* < 0.05, ** *p* < 0.01, *** *p* < 0.001.

**Figure 4 cancers-13-00872-f004:**
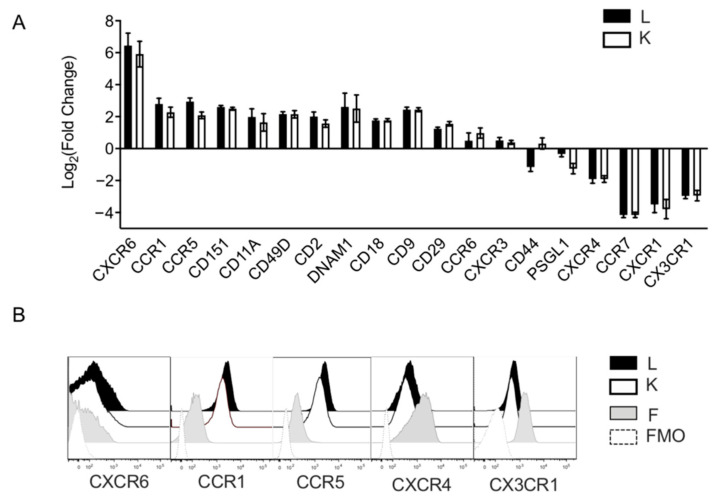
Ex vivo expansion with feeder cells induces a chemotactic receptor and adhesion molecule expression shift on NK cells. (**A**) Surface expression phenotype on expanded NK cells, compared to non-expanded NK cells. Data is reported as fold change of relative geometric mean florescent intensities (rGMFI). Black bars represent surface expression on NK cells that were expanded with EBV-LCL feeder cells and white bars represent surface expression on NK cells that were expanded with GE-K562 feeder cells. Error bars report the SEM and significance was analyzed with a paired *t*-test (*n* = 6 donors). (**B**) Histograms that represent surface expression levels of chemotactic receptors.

**Figure 5 cancers-13-00872-f005:**
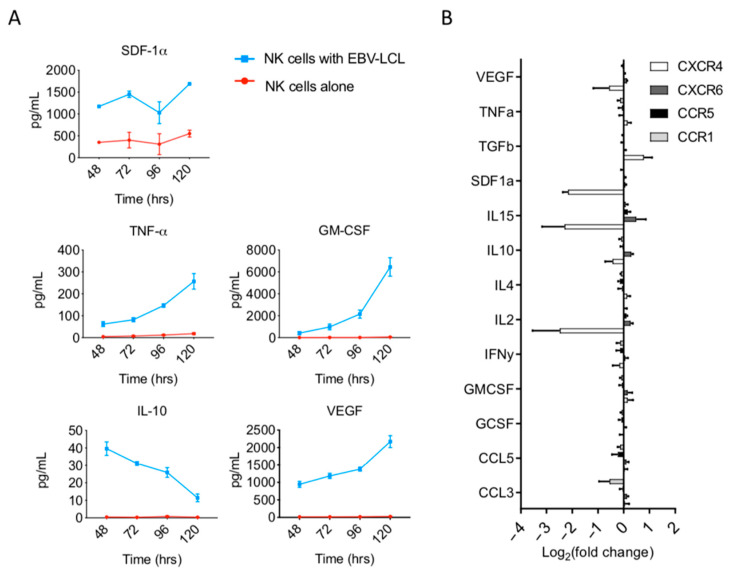
Ex vivo expansion of NK cells with EBV-LCL feeder cells creates a pro-inflammatory environment. (**A**) NK cell culture supernatants were analyzed using a multiplex assay for the presence of various cytokines and chemokines. Molecule concentrations are reported as pg/mL over time. Line graphs show data from supernatants collected from NK cell expansion with EBV-LCL feeder cells (blue) or from supernatants of NK cell alone in IL-2-containing media (red) (*n* = 2 technical replicates, 2 biological replicates). (**B**) Surface expression changes of chemotactic receptors in response to exposure to individual various cytokines and chemokines for 24 h in vitro (*n* = 3).

**Figure 6 cancers-13-00872-f006:**
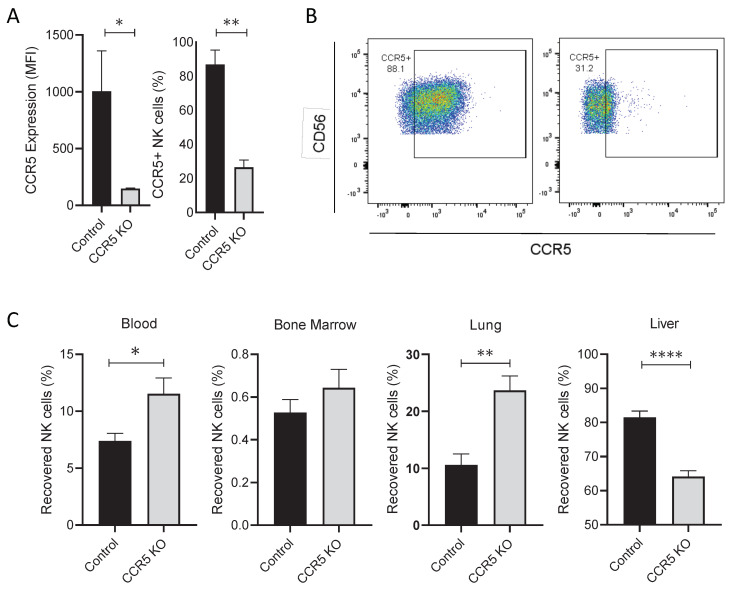
Reduction of CCR5 expression significantly reduces NK cell trafficking into the liver 24 h following NK cell infusion into mice. (**A**) Bar graphs that illustrate CCR5 positivity and surface expression on ex vivo expanded NK cells (*n* = 4 donors). (**B**) Dot plots representing CCR5 surface expression on NK cells (one representative donor). (**C**) Plots displaying the fraction of NK cells that were recovered in tissues, 24 h following i.v. infusion into mice (*n* = 8–9 mice per group, 2 independent experiments). Statistics determined with the Students *t*-test, two tailed, * *p* < 0.05, ** *p* < 0.01, **** *p* < 0.001.

## Data Availability

Deposition of RNA sequencing data can be accessed through the Gene Expression Omnibus database (accession GSE165498).

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
