# Peer review of "RNA-Seq Analysis Reveals CCR5 as a Key Target for CRISPR Gene Editing to Regulate In Vivo NK Cell Trafficking"

_cancers, 2021, doi:10.3390/cancers13040872_

Round 1

Reviewer 1 Report

This is an important manuscript, which highlights an important aspect of cellular immunotherapy, which is often overlooked, the ability of adoptively transferred cells to home to their desired destination.  The authors have previously shown in in-vivo models that a large number of NK cells end up in the liver following adoptive transfer.  The authors have now shown very convincing data that during NK cell expansion employing widely used feeder cell layers, that chemokine receptor expression on NK cells changes significantly with down regulation of receptors such as CXCR4 and up regulation of receptors including CCR5, which explains trafficking into the liver.  Deletion of this receptor, using gene editing with CRISPR/Cas9 significantly reduces the number of liver resident cells following adoptive transfer of expanded human NK cells to immunodeficient mice.  This could have important implications for future trials with adoptively transferred expanded NK cells with the potential for more efficient homing of CCR5 KO NK cells to sites of tumor, bypassing the liver.

Overall this was a very well written paper.  I have several minor comments:

Line 83: PACT should be spelt out as not obvious what this is to those outside NIH

Line 147: CRISPR not CIRSPR

Line 424:  The authors saw substantial changes in CD62L, JAK2, GRK6, and HIF-1 alpha gene expression.  Did they see any changes in PIM1 expression?  PIM1 is known to regulate CXCR4 via phosphorylation

Grundler R, Brault L, Gasser C, Bullock AN, Dechow T, Woetzel S et al. Dissection of PIM serine/threonine kinases in FLT3-ITD-induced leukemogenesis reveals PIM1 as regulator of CXCL12-CXCR4-mediated homing and migration. J Exp Med 2009; 206: 1957–1970.

Other comments:  The expression of CD44 and PSGL-1, both of which can act as scaffolds for E-selectin ligands, were reduced on expanded NK cells.  Did the authors consider the possibility that there could be changes in genes involved in selectin ligand synthesis e.g. ST3GAL4/ST3GAL6 and/or FUT6/7, which could result in alterations in expression of E-selectin ligands, which would also impact on trafficking into the bone marrow?

Author Response

Line 83: PACT should be spelt out as not obvious what this is to those outside NIH

  • This has been corrected, thank you.

Line 147: CRISPR not CIRSPR

  • This has been corrected, thank you.

Line 424:  The authors saw substantial changes in CD62L, JAK2, GRK6, and HIF-1 alpha gene expression.  Did they see any changes in PIM1 expression?  PIM1 is known to regulate CXCR4 via phosphorylation

  • This is a great suggestion! We checked for PIM1 expression, along with additional regulators of CXCR4 in our dataset. We modified figure 3c to include these regulators. This updated figure is discussed on lines 341 and 534 of the text.

The expression of CD44 and PSGL-1, both of which can act as scaffolds for E-selectin ligands, were reduced on expanded NK cells.  Did the authors consider the possibility that there could be changes in genes involved in selectin ligand synthesis e.g. ST3GAL4/ST3GAL6 and/or FUT6/7, which could result in alterations in expression of E-selectin ligands, which would also impact on trafficking into the bone marrow

  • Thank you for this comment. Yes, the ability for cells to express these proteins may have an impact on cellular trafficking. We do have data on the transcription of these glycoproteins and scaffold synthesis proteins, however this is not included in this manuscript, as this story is focused on chemokine receptors. Future studies will assess the importance and contribution of glycoprotein biology to the homing of adoptively transferred NK cells.

Reviewer 2 Report

Dr. Levy and colleagues in this elegant study extensively evaluated gene expression of fresh vs ex vivo expanded NK cells. They found major effects on the expression of several key genes in particular CXCR4 and CCR5 both quite relevant to the ongoing NK cell adoptive transfer efforts. Overall it is a nicely done study, however I have follow comments / suggestions (required) for the authors:

  1. Even though the CCR5 expression would favor 'trapping' of these cells in the liver, however it is not clear if this will impact in vivo tumor control, please demonstrate clinical relevance of this finding by showing impaired (and rescue by crispr edited cells) in vivo tumor control in xenograft mice using luc+ cell line like K562 or some other human tumor cell lines and or PDX
  2. Downregulation of CXCR4  is also very clinically relevant, I suggest knocking in / forcing expression of this gene and then again looking at the differential trafficking and in vivo tumor control.
  3. Would CCR5 over expression be potentially advantageous to control cholangiocarcinoma/ hepatic tumors? 
  4. Consider adding maraviroc to demonstrate an effect similar/or synergistic to the crispr edited CCR5 as maraviroc commercially available and an FDA approved drug. 

Author Response

Even though the CCR5 expression would favor 'trapping' of these cells in the liver, however it is not clear if this will impact in vivo tumor control, please demonstrate clinical relevance of this finding by showing impaired (and rescue by crispr edited cells) in vivo tumor control in xenograft mice using luc+ cell line like K562 or some other human tumor cell lines and or PDX

  • Thank you for this comment. Of course, tumor control is a salient point for highlighting the clinical relevance of therapeutic cell product. However, our ability to conduct and optimize these studies right now is limited due to COVID-related restrictions in the lab. The focus of this manuscript is cellular trafficking, and in future studies we will investigate tumor control.

Downregulation of CXCR4 is also very clinically relevant, I suggest knocking in / forcing expression of this gene and then again looking at the differential trafficking and in vivo tumor control.

  • We appreciate the reviewer’s comments. Indeed, we have previously published that genetic upregulation of CXCR4 or CCR7 using mRNA transfection of ex vivo expanded NK cells can transiently modulate the in vivo homing of adoptive infused modified NK cells (Carlsten, Front. Immun., 2016; Levy, Front Immun., 2019). We are currently optimizing a CRISPR knock out/knock in model as a separate project in the lab to induce a more sustained expression of the desired chemokine receptor. Once we have this model optimized, we will conduct studies that assess KO of various chemokine receptors (CCR5/ CCR1/ CXCR3/ CXCR6) with knock in of CXCR4 or CCR7. This is designed as a study that will complement but is separate from this manuscript.

Would CCR5 over expression be potentially advantageous to control cholangiocarcinoma/ hepatic tumors?

  • This is a very important question to consider. We have included our thought on this matter in the discussion section:

“Although adoptively transferred immune cells accumulate in the liver, it is not clear if they would be effective at controlling hepatic tumors. The liver tumor microenvironment is known to be particularly suppressive to hepatic NK cells, and would be predicted to have the same suppressive effects to adoptively transferred ex vivo expanded NK cells. Factors that contribute to the NK cell suppression by liver tumors include hypoxia, tumor production of metabolites such as lactate and adenosine (Husain et al.,2013, Young et al. 2018), tumor secretion of IL-10 and TGFb (Sun et al. 2017, Hequin et al. 2009), and IDO and PGE2 released by tumor-derived fibroblasts (Li et al. 2012). These factors are characteristic of hepatic tumors that can induce NK cell dysfunction or anergy“

Consider adding maraviroc to demonstrate an effect similar/or synergistic to the Crispr edited CCR5 as maraviroc commercially available and an FDA approved drug.

  • Thank you for this suggestion. We subsequently performed experiments using Maraviroc to pharmacologically inhibit CCR5. Unfortunately, the analysis from these experiments (below) proved to be inconclusive.

Mice were treated with 10mg/kg Maraviroc or PBS, followed by NK cell infusion using either Control NK cells or CCR5 KO NK cells. Twenty four hours after cell infusion, Liver, lungs, blood, and BM were harvested. NK cells were identified by flow cytometry. NK cells were identified as CD56+ cells in the size-selected, single, live cell population within each tissue.

When we evaluated the frequency of CD56+ cells in the liver, it appeared mice treated with Maraviroc had a reduction in the trafficking of control NK cells into the liver (figure a). However, when we normalized this frequency relative to the total number of NK cells that we recovered from all tissues per mouse, we did not observe a significant difference between NK cells in the liver of Maraviroc-treated vs. non-treated mice (figure b). The correction that this normalization is making could be attributed to the viability of control NK cells in mice that have been treated with maraviroc. Thus, we believe additional experiments would need to be conducted to thoroughly evaluate the effect of Maraviroc on NK cells. We included a statement in our discussion to address this “Future studies will evaluate the potential of pharmacological inhibition of CCR5 to reduce the trafficking of infused cells into liver tissue.”

Round 2

Reviewer 2 Report

The authors have appropriately addressed my comments.